# Ketoprofen-Based Polymer-Drug Nanoparticles Provide Anti-Inflammatory Properties to HA/Collagen Hydrogels

**DOI:** 10.3390/jfb14030160

**Published:** 2023-03-17

**Authors:** Norbert Halfter, Eva Espinosa-Cano, Gloria María Pontes-Quero, Rosa Ana Ramírez-Jiménez, Christiane Heinemann, Stephanie Möller, Matthias Schnabelrauch, Hans-Peter Wiesmann, Vera Hintze, Maria Rosa Aguilar

**Affiliations:** 1Institute of Materials Science, Max Bergmann Center of Biomaterials, Technische Universität Dresden, Budapester Straße 27, 01069 Dresden, Germany; 2Group of Biomaterials, Institute of Polymer Science and Technology ICTP-CSIC, C/Juan de la Cierva 3, 28006 Madrid, Spain; 3CIBER de Bioingeniería, Biomateriales y Nanomedicina, Instituto de Salud Carlos III, C/Monforte de Lemos 3/5, 28029 Madrid, Spain; 4Department of Biomaterials, INNOVENT e. V., Prüssingstraße 27B, 07745 Jena, Germany

**Keywords:** ketoprofen, nanoparticles, hyaluronan, collagen, hydrogels, cryogels, anti-inflammatory, macrophages, wound dressing

## Abstract

Current limitations of wound dressings for treating chronic wounds require the development of novel approaches. One of these is the immune-centered approach, which aims to restore the pro-regenerative and anti-inflammatory properties of macrophages. Under inflammatory conditions, ketoprofen nanoparticles (KT NPs) can reduce pro-inflammatory markers of macrophages and increase anti-inflammatory cytokines. To assess their suitability as part of wound dressings, these NPs were combined with hyaluronan (HA)/collagen-based hydro- (HGs) and cryogels (CGs). Different HA and NP concentrations and loading techniques for NP incorporation were used. The NP release, gel morphology, and mechanical properties were studied. Generally, colonialization of the gels with macrophages resulted in high cell viability and proliferation. Furthermore, direct contact of the NPs to the cells reduced the level of nitric oxide (NO). The formation of multinucleated cells on the gels was low and further decreased by the NPs. For the HGs that produced the highest reduction in NO, extended ELISA studies showed reduced levels of the pro-inflammatory markers PGE2, IL-12 p40, TNF-α, and IL-6. Thus, HA/collagen-based gels containing KT NPs may represent a novel therapeutic approach for treating chronic wounds. Whether effects observed in vitro translate into a favorable profile on skin regeneration in vivo will require rigorous testing.

## 1. Introduction

In Western civilizations, there is an increasing number of chronic skin wounds as the population ages [1]. The consequences are a reduced quality of life and high costs for the health care system. Diseases, such as diabetes and obesity, cause a further rise in chronic skin wounds due to detrimental pre-existing conditions, leading to a higher likelihood of treatment failure and recurrence, which further increases discomfort and costs [2]. Although various therapies are available for chronic wound treatment and care, their effectiveness is somewhat limited, and relapse occurs in many patients [2,3,4]. As a result, the development of novel materials for improved treatment of chronic skin wounds is needed.

One promising strategy for promoting the healing chronic wounds is an immune-centered approach [5]. Here, the mechanisms of the immune system are to be modulated for healing to occur. Besides danger signals from damaged cells, neutrophils and macrophages play an important role in this respect. In chronic wounds, pro-inflammatory M1-like macrophages are persistent, resulting in prolonged inflammation [6]. However, restoring pro-regenerative and anti-inflammatory M2-like macrophages can help to improve the healing process [7,8].

For the treatment of chronic wounds, films, foams, hydrocolloids, and hydrogels (HGs) are used as dressings that are made from polyurethanes, alginate, collagen (coll), pectin, carboxymethyl cellulose, and propylene glycol [9,10]. All of them create the moisture balance that allows wounds to heal faster. Depending on the material used, other beneficial properties for wound healing may also be present. There are a large number of wound dressings from various manufacturers that fall into the abovementioned material categories [3,9,10,11,12,13,14,15,16,17]. However, studies in the clinic show that repair via scar formation predominates, with no physiological regeneration of the skin [18]. Furthermore, adhesive dressings can take off a large portion of healthy or newly formed skin during removal [19]. In addition, extensive clinical studies to evaluate the effectiveness of available wound dressings are limited, and the superiority of particular wound dressings is indistinguishable [20].

In the past, components of the extracellular matrix (ECM) have proven to be suitable building blocks for an immune-centered approach, as they are involved in the process of wound healing through interaction with cells, growth factors, and cytokines [21,22,23,24,25,26]. One major component of the ECM of skin is hyaluronan (HA), a glycosaminoglycan composed of repeating disaccharide units of D-glucuronic acid and N-acetyl-D-glucosamine [(-4GlcUAβ1-3GlcNAcβ1-)n]. The role of HA in wound healing is emphasized by its involvement in all stages of wound healing, e.g., blood clotting, cell migration and infiltration, as well as inhibition of neutrophil migration, gap filling, and structural organization for the newly forming ECM, as summarized elsewhere [8,27,28].

HA-based wound dressings are also available commercially but face the same problems as other dressings. Generally, there are only a small number of participants in clinical trials, and the dressing may be insufficient for the complex real-life situations that the clinicians have to handle (e.g., different ages, depth of the wound, and wound position). Nevertheless, Gravante et al. show the potential of HA dressings, as 74% of all patients achieved wound closure within 39 days [29]. However, at least 7% of the patients showed only partial re-epithelialization after the same time period, which makes further improvements for these dressings desirable. 

In vitro studies have shown that HA/coll-based HGs were a suitable growth environment for endothelial cells, keratinocytes, and fibroblasts [30,31]. A recent study demonstrated that the same HGs, releasing chemically sulfated HA (sHA) as an immunoregulatory component, improved tissue repair in chronic wounds of diabetic db/db mice, i.e., by enhancing pro-regenerative M2-macrophage activation [32].

Compared to the HGs without sHA, wound closure increased from 30% to 50% of the cases investigated. Although this is already a marked improvement, the results still indicate that further measures need to be taken, which could be the combination with further anti-inflammatory agents.

The use of non-steroidal anti-inflammatory drugs (NSAIDs) offers an additional anti-inflammatory treatment by influencing macrophage polarity and improving the healing of chronic wounds [33,34]. The NSAIDs are effective because of their inhibition of cyclooxygenase-2 (COX-2), which is responsible for the synthesis of prostaglandins (PG) in pathological processes, such as inflammation [35]. Furthermore, NSAIDs have COX-independent mechanisms of action that contribute to their anti-inflammatory properties [36]. Unfortunately, most NSAIDs are hydrophobic, which results in low bioavailability. To address this problem, hydrophilic nanoparticles (NPs) with NSAID moieties have been developed [33,34]. Previously, in vitro experiments with ketoprofen (KT)-based NPs and macrophages showed low toxicity, as well as a reduction in nitric oxide (NO) and other pro-inflammatory markers, such as Il-12b and TNF-α in inflammatory conditions [34]. In addition, anti-inflammatory markers, such as VEGF and Il-10, were increased in expression. These results suggest that NSAID-containing NPs are a promising immunoregulatory component to be included in functional wound dressings for regulating macrophage activities in non-healing wounds.

In this study, we combined HA/coll-based HGs and CGs with anti-inflammatory KP NPs for the first time. The aim was to assess the NP’s suitability in enhancing the anti-inflammatory properties of these HGs and, thus, to reveal the composite’s potential for application as an improved anti-inflammatory wound dressing. In this context, NP concentrations and loading technique for NP incorporation was expected to influence the efficacy of NPs. Thus, we designed different HA/coll-based HG and CG variants with NSAID-containing NPs. The loading capacity and NP distribution within the gels, NP release, HG and CG morphology, and mechanical properties were determined. Furthermore, cell viability, proliferation, NO release, multinucleated giant cell formation, and the production of inflammatory and anti-inflammatory markers were used to assess the cellular response of a murine macrophage cell line to NP-loaded gels. The outcome of this study was expected to clarify which gel variant and NP concentration will be the most suitable for prospective combinatory approaches with other anti-inflammatory compounds and in vivo experiments in an animal model for chronic skin wounds.

## 2. Materials and Methods

### 2.1. Hyaluronan Methacrylate Preparation

For HA methacrylate (HA-MAC) synthesis, native HA (MW = 1400 kDa from Streptococcus, Kraeber & Co GmbH, Ellerbek, Germany) was processed as reported before [37]. In brief, HA was dissolved in borate buffer (pH 8.5, Sigma-Aldrich, Taufkirchen, Germany) and treated with a 15-fold molar excess of methacrylic acid anhydride (Sigma-Aldrich, Taufkirchen, Germany) at 5 °C for 24 h. To purify the product, it was precipitated in acetone (Sigma-Aldrich, Taufkirchen, Germany) and dialyzed against water (MWCO = 3500 g/mol).

### 2.2. Nanoparticle Synthesis

NPs were synthesized using the previously described copolymer poly(HKT-co-VI) (48:52), obtained by free radical copolymerization of a methacrylic derivative of ketoprofen (HKT) and 1-vinylimidazole (VI) [34]. After analyzing the copolymer by ^1^H-NMR (Varian Mercury, 400 MHz, 25 °C), the NPs were obtained by the nanoprecipitation method (Figure 1). Briefly, an organic solution (acetone (Scharlau, Barcelona, Spain): ethanol (Scharlau), 80:20 (V:V)) of the copolymer (10 mg/mL) was added dropwise to an aqueous buffer solution at pH 4 (0.1 M acetic acid and 0.15 M NaCl, both Panreac, Barcelona, Spain). The remaining organic solvent was eliminated by evaporation under continuous stirring overnight, and the resultant NPs were stored at 4 °C until used.

For binding and release studies and the visualization of the NPs in the HGs and CGs, the NPs were loaded with coumarin-6 (C6, Sigma-Aldrich, Lyon, France). For this, C6 (1% w:w with respect to the polymer) and the corresponding copolymer were dissolved in a mixture of acetone:ethanol (80:20, V:V) and slowly dropped into the aqueous buffer solution (0.1 M Acetic Acid, 0.1 M NaCl) at pH 4 under magnetic stirring. NPs were dialyzed against the same buffer for 72 h to eliminate remaining organic solvents and the soluble non-entrapped C6. The resultant NPs were filtered through 1 μm nylon filters (Whatman Puradisc, cytiva, Barcelona, Spain) to eliminate insoluble C6.

### 2.3. NP Characterization

The mean hydrodynamic diameter (D_h_) of NPs, size distribution, and polydispersity index (PDI) were studied by dynamic light scattering (DLS) using a Malvern Nanosizer Nano-ZS Instrument (Madrid, Spain) equipped with a 4 mW He–Ne laser (λ = 633 nm) at a scattering angle of 173°. The zeta potential (ξ) was determined by laser Doppler electrophoresis (LDE) using a Malvern Nanosizer Nano-ZS Instrument equipped with a 4 mW He-Ne laser at 25 °C. Experiments were performed in triplicate, and results were expressed as the statistical average ± standard deviation (SD).

### 2.4. Scaffold Fabrication

#### 2.4.1. Preparation of Hydro- and Cryogels

HGs were prepared as previously described [30]. Briefly, a 1 mg/mL coll solution was prepared by mixing rat tail coll type I (Corning, New York, NY, USA) and 0.01 M acetic acid (Sigma-Aldrich, Taufkirchen, Germany). Before in vitro fibrillogenesis at 37 °C for 4 h, the coll solution was diluted with fibrillogenesis buffer (0.05 M Na_2_HPO_4_, Carl-Roth, Karlsruhe, Germany and 0.01 M KH_2_PO_4_, pH 7.4, Sigma-Aldrich, Taufkirchen, Germany) to a concentration of 0.5 mg/mL coll. After fibrillogenesis, HA-MAC was dissolved in fibrillated coll solution to reach concentrations of 10 or 30 mg/mL HA-MAC, followed by the addition of 10 mg/mL lithium phenyl-2,4,6-trimethylbenzoyl-phosphinate (LAP, TCI Deutschland GmbH, Eschborn, Germany) in a ratio of 1:10 (V:V). Either 50 µL of this solution was pipetted between two cover slides (Ø = 12 mm, VWR, Darmstadt, Germany) coated with Sigmacote (Sigma-Aldrich, Taufkirchen, Germany), or 200 µL was pipetted into mold casts (Ø = 7.5 mm, just for porosity experiments). HGs were photo-crosslinked by UV irradiation (365 nm, 0.17 W/cm^2^, for 10 min), frozen for 30 min at −80 °C, and then freeze-dried with a Martin Christ Epsilon 2–4 LSC device (freeze-dry steps: (1) pre-cooling to −15 °C, (2) reducing pressure to 1.030 mbar at −15 °C for 105 min, (3) heating to 20 °C in 150 min under 1.030 mbar, (4) drying at 1.030 mbar and 20 °C for 720 min, (5) pressure reduction to 0.001 mbar and 20 °C in 10 min, (6) heating to 30 °C under 0.001 mbar in 50 min, and (7) drying at 30 °C and 0.001 mbar for 120 min). CGs were obtained similarly to HGs, except that they were frozen at −80 °C for 30 min before UV irradiation. More details about the fabricated gels can be found in Table 1.

#### 2.4.2. Loading of the Gels with NP

NPs were loaded into the gels via different methods, i.e., by soaking of freeze-dried scaffolds or by incorporation during the fabrication process. To determine the concentration of NPs needed to achieve a particular loading level, first loading studies were performed with 1 mg/mL NP solution (max NP; Table 1; Appendix A, Appendix A). Based on this, the adapted concentrations were used for subsequent loading of HGs and CGs with NPs (Table 1; Appendix A, Appendix A). In general, NP loading of HGs and CGs via soaking was carried out by incubating the scaffolds with 125 µL NP solution on the top and bottom sides separately for 24 h at RT to ensure even NP distribution. After soaking, the samples were washed twice with 500 µL phosphate buffered saline (PBS, Sigma-Aldrich, Taufkirchen, Germany). Subsequently, the gels were frozen for at least 30 min and freeze-dried before use. C6-loaded NPs were used for determining the NP release and concentration inside the gels, along with 3D visualization with confocal light scanning microscopy (CLSM). All HGs and CGs used for cell experiments were prepared from NPs without C6.

The incorporation of NPs into HGs during the preparation process was carried out by adding the NPs before the chemical crosslinking reaction (CL-10HA 40NP sample, Table 1). Here, the procedure in Section 2.4.1. was followed, except for the step of dissolving the HA-MAC, which was performed in 0.5 mg/mL coll containing 0.8 mg/mL NPs. For this purpose, coll was diluted with deionized water and mixed with 10× fibrillogenesis buffer (0.5 M Na_2_HPO_4_, 0.1 M KH_2_PO_4_, pH 7.4) in a ratio of 1:9 (V:V) to obtain a 2.5 mg/mL coll solution. This was fibrillated first for 4 h and subsequently mixed with 1 mg/mL NP solution in a ratio of 1:4 (V:V) followed by dissolving of HA in this solution. After adding 1/10th of the volume of 10 mg/mL LAP, 50 µL of the mixture was pipetted between two cover slides with subsequent UV crosslinking, freezing, and freeze-drying. Before use, CL-10HA 40NP and all other scaffolds not loaded with NPs by soaking were swollen in 500 µL PBS at RT for 1 h, washed twice with 500 µL PBS, and freeze-dried again. An overview of the different fabrication procedures can be found in Figure 2.

### 2.5. NP Release Studies from Hydro- and Cryogels

Those HGs and CGs loaded with 1 mg/mL NPs (max NP) and subsequently washed with PBS were incubated in 500 µL PBS for either 1 d or 7 d at 37 °C. Gels for the NP release studies were used directly after washing without subsequent freeze-drying. Washing solutions, supernatants, and gels were stored at −20 °C prior to analysis. 

### 2.6. Determination of NP Concentration

The NP concentration in washing solutions, supernatants, and gels was determined by measuring the fluorescence (Tecan, Infinite^®^ M200 PRO, λ_Exc_ = 485 nm, λ_Emi_ = 528 nm). Gels were digested with 600 µL of 0.01 M acetate buffer (pH 5.35) containing 0.15 M NaCl (Sigma-Aldrich, Taufkirchen, Germany) and 1000 U/mL hyaluronidase (HYAL, EC 3.2.1.35, from bovine testes, Sigma-Aldrich, Taufkirchen, Germany) prior to the determination of the NP concentration (Hauck et al. 2021). For more details about the digestion conditions, see Table 2. Since CGs could not be digested completely, the remaining CGs were extracted twice with 150 µL EtOH. After the extraction, EtOH supernatants were centrifuged and measured.

Prior to the determination of NP concentration, acidic solutions containing fluorescent NPs (NP-stock solution and digestion solutions) were made basic by adding 1/10th volume of 1 M NaOH. Subsequently, all solutions containing NPs were centrifuged at 10,000× *g* RCF for 5 min, and the supernatants were discarded. Next, pellets (NPs) were resuspended in 150 µL ethanol absolute (EtOH, VWR, Fontenay-sous-Bois, France), and centrifuged again at 10,000× *g* RCF for 5 min before the fluorescence was measured. In the next step, the obtained NP amounts were used to calculate the NP concentration incorporated in the gels.

### 2.7. NP Distribution in Hydro- and Cryogels

CLSM was used for the visualization of NPs in gels. For this purpose, freeze-dried samples were swollen for up to 30 min in PBS and measured on an upright Axioscop 2 FS mot microscope equipped with a LSM 510 META module (Zeiss, Jena, Germany) using an argon+ laser for excitation of C6 at 488 nm.

### 2.8. Hydro- and Cryogel Morphology

Scanning electron microscopy (SEM, Philips ESEM XL 30, FEI) using a secondary electron detector was used to assess the morphology of freeze-dried gels. The HG and CG scaffolds were cut with a scalpel and mounted on the SEM stage, followed by sputter coating (MED 010, Balzers, Balzers, Liechtenstein) with carbon (Plano, Wetzlar, Germany). The images were taken at different magnifications using an acceleration voltage of 3 kV and a spot size of 3.

### 2.9. Mechanical Characterization of the Hydro- and Cryogels

The elastic modulus of the swollen samples was determined using the CellScale Microsquisher (Waterloo, ON, Canada). After swelling freeze-dried samples for 1 h in PBS, uniform HGs and CGs with a 3 mm diameter were obtained with a biopsy punch. After measuring the height of the gels with the integrated digital camera, force–displacement curves were obtained at a velocity of 1.67 µm/s in a range below 15% displacement. The linear area of the stress–strain curve was used to determine the elastic modulus. To reveal significant differences, one-way ANOVA was performed for *p* < 0.05 (*n* = 3).

### 2.10. Porosity of the Hydro- and Cryogels

The porosity (*Φ*) was determined with a pycnometer and calculated using the following formula:(1)$$\Phi =1-\frac{{(m}_{1}-m_{2}+m_{3})}{V_{T}\cdot P_{PBS}}$$
where m_1_ is the mass of the pycnometer filled with PBS, m_2_ is the mass of the sample in the pycnometer filled with PBS, m_3_ is the mass of the freeze-dried sample, Ρ_PBS_ is the density of PBS (1.0047 g/mL) at RT (22.5 °C), and V_T_ is the total volume of the swollen sample. V_T_ was determined using the dimensions of the cylindrical 200 µL samples (HGs using a caliper, and CGs using the Microsquisher).

### 2.11. Cell Culture

Cells were cultured using a murine macrophage cell line (RAW264.7, 91062702, Sigma-Aldrich, Gillingham, UK). Cells were maintained over permissive conditions in high-glucose Dulbecco’s modified Eagle’s medium (DMEM; D6546, Sigma, St. Louis, MO, USA) supplemented with 10% fetal bovine serum (FBS; Gibco, Brazil, Thermo Fisher, Madrid, Spain), 2% L-glutamine (2 × 10^−3^ M, Sigma, St. Louis, MO, USA), and 1% penicillin G (100 U/mL, Sigma, St. Louis, MO, USA) at 37 °C in a humidified incubator with 5% CO_2_.

### 2.12. Direct Assays

#### 2.12.1. Micro-Mass Cell Seeding

Before cell seeding, HGs and CGs were pre-incubated with 500 µL of complete RAW264.7 culture media in 24-well plates for 1 h at 37 °C. Then, 50 µL of RAW264.7 (2 × 10^6^ cells/mL) were micro-mass seeded on top of each gel (i.e., 10HA, 10HA 40NP, 10HA 120NP, CL-10HA 40NP, 30HA, 30HA 40NP, cryo 10HA, and cryo 10HA 40NP). After 40 min of incubation under permissive conditions (37 °C, 5% CO_2_), 500 µL of culture media were added to each well, and cells were incubated overnight.

#### 2.12.2. Cell Proliferation

Cell proliferation was determined in the HGs and CGs using an AlamarBlue^®^ (Bio-Rad Laboratories, Inc., manufactured by Trek Diagnostic System, Hercules, CA, USA) assay after 24 and 48 h. All gels were incubated with the AlamarBlue^®^ reagent at 10% (V:V) in medium without phenol red at 37 °C for 3 h. Fluorescence of reduced AlamarBlue^®^ was determined at 530/590 nm excitation/emission wavelengths (Synergy HT, BIO-TEK, Winooski, VT, USA). ANOVA was performed at a significance level of *p* < 0.05.

#### 2.12.3. Cell Viability

Cell viability was determined after 48 h of incubation using a Live/Dead™ Viability/Cytotoxicity Kit (Invitrogen Inc., Grand Island, NY, USA). All HGs and CGs were incubated with PBS containing Calcein AM (2 µM, Sigma-Aldrich, Madrid, Spain) and ethidium homodimer (4 µM, Sigma-Aldrich, Madrid, Spain) at 37 °C for 30 min to stain live and dead cells, respectively. Gels were imaged with an inverted microscope (4-fold magnification, Nikon Eclipse TE 2000-S, Tokio, Japan) and analyzed using Image J software [38]. Treatments were carried out in triplicates, and five different images of each replicate were analyzed. The percentage of viable cells is given as stated in Equation (2). ANOVA was performed at a significance level of *p* < 0.05.
(2)$$Viable\;cells\left[\%\right]=\frac{viable\;cells}{total\;cells}\times 100$$

### 2.13. Anti-Inflammatory Activity of Hydrogels

#### 2.13.1. Nitric Oxide Production

Lipopolysaccharide (LPS)-induced NO release was evaluated using a Griess reagent kit (Sigma-Aldrich, St. Louis, MO, USA). Briefly, after 24 h on top of the gels, cells were treated with LPS (500 ng/mL; from Escherichia coli O111:B4; CAS Number: 297-473-0, Aldrich) and incubated overnight under permissive conditions. Then, medium from each well was collected, and NO released by RAW264.7 was determined using a Griess test according to the manufacturer’s instructions. Briefly, 75 µL of collected medium from each sample was mixed with 25 µL of Griess reagent and incubated in the dark at RT for 15 min. Then, absorbance at 540 nm was measured by a microplate reader. Treatments were carried out using four replicates. LPS-stimulated cells seeded directly on the well plate were used as a control, taking this result as the 100% NO production (Equation (3)). ANOVA was performed at a significance level of *p* < 0.05.
(3)$$NO\;released\left[\%\right]=\frac{NO\;released\;(LPS)-activated\;cells\;(hydrogels)}{NO\;released\;(LPS)-activated\;cells\;(well\;plate)}\times 100$$

#### 2.13.2. Macrophages Spreading and Multinucleated Cells Evaluation by Fluorescence Microscopy

HG and CG effect on RAW264.7 macrophage spreading and multinucleated cell formation after 48 h of incubation was investigated by fluorescence microscopy of actin filaments and nuclei. Macrophages attached to the surface, in the presence or absence of LPS (500 ng/mL), were fixed for 30 min with 37% (V:V) paraformaldehyde (Sigma-Aldirch, Madrid, Spain) in distilled water. This was followed by permeabilization with 0.05% (V:V) Triton X-100 (Sigma-Aldirch, Madrid, Spain) and washing with distilled water. Actin filaments and nuclei were stained with 10 ng/mL rhodamine phalloidin (red) (Invitrogen, Thermo Fisher Scientific, Waltham, MA, USA) diluted 1:1000 with PBS, and Hoechst 33342 fluorescent dye (blue) (Invitrogen, Thermo Fisher Scientific, 2161855, Waltham, MA, USA) diluted 1:100 with PBS, respectively, for 30 min at RT. After repeated washing steps with distilled water, samples were incubated with 200 µL of Tween 20 (10%w) for 5 min, washed with distilled water, and examined using an inverted microscope (20-fold magnification, Nikon Eclipse TE 2000-S). The macrophage surface area was quantified in µm^2^ using Image J software [38]. For statistical purposes, samples were evaluated in duplicates, 5–10 photos were taken, and the area of 20–50 cells was measured per sample.

#### 2.13.3. ELISA for Anti- and Pro-Inflammatory Markers

To assess the anti-inflammatory effect of 10HA and 10HA 120NP HGs further, ELISA assays were performed to quantify the release of six pro-inflammatory mediators by LPS-stimulated RAW264.7. Mouse TNF-α, IL-6, IL-10, IL-12p40, IL-23, and PGE2 ELISA kits were purchased from abcam (ab208348, ab222503, ab255729, ab236717, ab119545, and ab133021, Cambridge UK). In brief, cells were seeded on top of the HGs and CGs at a density of 2 × 10^6^ cells/well and cultured for 24 h. Then, cells were exposed to medium containing LPS (500 ng/mL) and incubated overnight. Media were collected and stored at −20 °C until use. Levels of mouse TNF-α, IL-6, IL-10, IL-12p40, IL-23, and PGE2 in cell culture supernatants were determined by the corresponding ELISA kit according to the protocol recommended by the manufacturer. Experiments were performed using six replicates per formulation. ANOVA was performed at significance levels of *p* < 0.05, *p* < 0.01, and *p* < 0.001.

### 2.14. Indirect Assays

#### 2.14.1. Extract Collection from Hydrogels

Cell proliferation and NO production were also assessed using indirect assays. Here, 10HA, 10HA 40NP, 30HA, 30HA 40NP, cryo 10HA, and cryo 10HA 40NP HGs were placed in Eppendorf tubes with 3 mL of complete RAW264.7 culture medium and kept at 37 °C under shaking. After 1, 2, 7, and 14 days, extracts were collected (3 mL) and stored at −80 °C. The volume of each tube was refilled with fresh medium. Three replicates were used per HG.

#### 2.14.2. Indirect Cell Proliferation Assay

RAW264.7 proliferation, when exposed to the extracts, was performed through an indirect test using the AlamarBlue^®^ reagent. First, RAW264.7 cells were seeded in 96-well plates at a concentration of 2 × 10^5^ cells/mL and incubated for 24 h at 37 °C and 5% CO_2_. Then, the medium was removed, and 100 μL of the previous extract was added to each well and incubated overnight. Finally, the extracts were removed, and 100 μL of AlamarBlue^®^ reagent at 10% (V/V) in medium without phenol red were added per well and incubated for 3 h. The fluorescence intensity of reduced AlamarBlue^®^ was determined at 530/590 nm excitation/emission wavelengths. ANOVA was performed at a significance level of *p* < 0.05.

#### 2.14.3. Indirect Nitric Oxide (NO) Assay

LPS-induced NO release was evaluated using a Griess reagent kit. In brief, cells were seeded into 96-wellplates at a concentration of 2 × 10^6^ cells/mL and incubated for 24 h at 37 °C and 5% CO_2_. Then, the medium was removed and 100 μL of the gel extracts and LPS (500 ng/mL) were added to each well and incubated overnight. The medium from each well was collected and the NO released was determined using a Griess test. Briefly, 75 µL of collected medium from each sample was mixed with 25 µL of Griess reagent and incubated in the dark at RT for 15 min. Then, absorbance at 540 nm was measured by a microplate reader. Treatments were carried out using four replicates. LPS-stimulated cells without extract exposure were used as a control. ANOVA was performed at a significance level of *p* < 0.05.

### 2.15. Statistical Analysis

The results are given as mean ± SD if not otherwise stated. One- or two-way analysis of variance (ANOVA) with additional Tukey’s multiple comparison tests, one sample, and unpaired *t*-test were considered significant for *p* values < 0.05. For statistics of the NP release experiment with an uneven number of samples, Mood’s median test was performed at a significance level of *p* < 0.05. In biological assays, one-way ANOVA between controls was labeled with * *p* < 0.05, ** *p* < 0.01, *** *p* < 0.001, and between samples with # *p* < 0.05.

## 3. Results and Discussion

### 3.1. Hydrodynamic Characterization of Ketoprofen NP

The synthesis of the copolymer poly(HKT-co-VI) was successful and was confirmed by ^1^H-NMR. KT NPs were successfully fabricated owing to the appropriate hydrophobic–hydrophilic balance of the copolymer poly(HKT-co-VI) that allows its self-assembly by the nanoprecipitation method [34]. As a consequence, NPs consist of a hydrophobic core made of covalently linked KT and a hydrophilic shell based on VI. NPs showed unimodal size distribution via light scattering, with a polydispersity index (PDI) below 0.1, a mean hydrodynamic diameter (D_h_) of 161 nm, and a positive surface charge of +30 mV (Figure 3A). These results are in agreement with the ones obtained in previous work, where the suitability of these hydrodynamic properties for macrophage internalization was demonstrated [34].

### 3.2. Binding and Release of Ketoprofen NP from HA/Coll Hydro- and Cryogels

Controlling the NP concentration in the different HG types is necessary for studying dose-dependent effects. To determine the concentration of NPs needed to achieve a particular loading level, loading studies were performed via soaking with a 1 mg/mL NP solution (max NP) and incubating in PBS at 37 °C for one and seven days. Experiments revealed no significant differences in NP concentration between variants and time points (Figure 3B). However, there is a clear tendency for a lower NP content in 30HA HGs compared to cryo 10HA gels after one and seven days (~50 µg NP/gel compared to ~100 µg/gel). The NP concentration in the gels was determined after sample digestion with hyaluronidase (HYAL; 37 °C, pH 5.35). While digestion of 10HA and 30HA HGs was completed in two to six days, CGs could not be completely digested within this period. This means that the NP amount regarding the CGs cannot be accurately determined. Thus, there could be more NPs in the CGs than are actually shown in Figure 3B,D. The denser HA network in 30HA resulted in a prolonged digestion time by the enzyme compared to the 10HA HG. However, the incomplete digestion of the cryo 10HA samples within the analyzed time suggests a structural difference. Since the SEM images did not reveal a substantial difference between 10HA and cryo 10HA samples on the microscopic scale, the structural difference might be found on the molecular scale (Figure 5 and Appendix A). This might be due to the cryoconcentration effect. Generally, the freezing of the solvent led to the increased concentration of the precursor and the photoinitiator, resulting in denser pore walls and possibly a higher degree of crosslinking [39,40]. Additionally, the cryoconcentration could increase the physical entanglement of HA chains resulting in a stronger resistance against HYAL, as shown by Cai et al. in vitro and in vivo, comparing HA HGs made by a cryo-procedure with non-cryogenic HA HGs [41,42].

In line with the direct determination of the NP content in the gels, the NP release, determined in the supernatants over seven days, suggested a strong binding of the NPs with almost no release after one day in PBS. While there was some release from 10HA and 30HA HGs after seven days, there was none from cryo 10HA over the investigated period (Figure 3C).

### 3.3. Controlled Loading of Hydro- and Cryogels with NP

In order to determine the most suitable NP concentration to be introduced into the HGs for a therapeutic effect, previous findings from Espinosa-Cano et al. were taken into account [34]. Here, NP concentrations ranging from 11 to 125 µg/mL significantly reduced NO production by LPS-stimulated RAW264.7. Consequently, 40 or 120 µg of NPs were added per HG. Figure 3D shows that the target concentration could be achieved for 30HA 40NP and 10HA 120NP HGs. However, 10HA 40NP, CL-10HA 40NP, and cryo 10HA 40NP HGs presented 23.8 ± 1.6, 11.8 ± 0.2, and 1.0 ± 0.5 µg NP/HG, respectively, resulting in statistically significant differences between the target concentration of 40 µg/HG and the obtained NP concentration. Nonetheless, the findings for loaded 10HA HG (40NP vs. 120NP) indicate that it is possible to control NP concentration in the HGs by choice of NP incubation concentration. For CL-10HA 40NP, it was assumed that all NPs were bound tightly in the organic matrix of the HGs and were not lost during the preparation procedure. However, the coumarin-6 (C6) bound to the NPs might have been destroyed during the photo-crosslinking procedure. As a result, lower fluorescence signals were detected, leading to a lower amount of calculated NPs. The very low amount of NPs found in the cryo 10HA 40NP samples might be related to the incomplete digestion of the CGs not releasing the NPs, or to the CG scaffold itself, holding back the fluorescence dye even after extraction with ethanol (Appendix A).

### 3.4. NP Distribution in Hydro- and Cryogels

Furthermore, CLSM (fluorescent C6-loaded NPs) and SEM were used to study the NP distribution in the HGs (Figure 4 and Figure 5). HGs without NPs (10HA, 30HA, and cryo 10HA) showed no fluorescence in CLSM or NPs in SEM (Figure 4A,G and Appendix A). The HGs loaded via soaking (10HA 40NP, 10HA 120NP, and 30HA 40NP) displayed NPs only on their surface, as observed with CLSM and SEM alike (Figure 4H,I,K, and Figure 5A,B,D). In contrast, the CL-10HA 40NP sample also displayed NPs in zones of at least 100 µm below the surface (Figure 4J). In SEM, however, the NPs were rarely visible on the surface but were rather embedded in the polymeric matrix due to the subsequent mixing of all components before crosslinking (Figure 5C,H). A similar distribution could be seen for the cryo 10HA 40NP sample, which is loaded via soaking. In this case, NP distribution in CLSM also reaches zones that are at least 100 µm below the surface. SEM showed NPs in the cross-section, demonstrating an open and interconnected porous structure where NPs can easily diffuse into the CGs (Figure 4L and Figure 5J). The large amount of NPs found via CLSM within cryo 10HA 40NP was in contrast to the determined NP content in the same samples, with only 1.0 ± 0.5 µg NP per gel (Figure 3D). These results confirm the aforementioned challenges in correctly determining NPs in CGs due to an incomplete HYAL digestion and ethanol extraction (see Section 3.3).

### 3.5. Morphology and Mechanical Properties of Hydro- and Cryogels

SEM images of the top view of all HGs showed a patterned but closed surface, while CGs displayed several open pores (Figure 5 and Appendix A). The cross-section of the freeze-dried gels revealed many pores, mostly smaller than 100 µm, and CG sample heights were higher in SEM than for HGs (Appendix A). Some of the HGs containing NPs shrank significantly during freeze-drying (Appendix A). However, there was no difference between the HG heights after swelling (Figure 6A). Similarly, freeze-dried CGs with NPs also displayed a decreased height in SEM compared to those without NPs. The loss in height after freeze drying can be attributed to the influence of salt residues from PBS and NPs, shielding the charge of carboxylic groups, allowing HA chains to move closer together. After swelling, the cryo 10HA samples were significantly higher than all other samples and the cryo 10HA 40NP samples were significantly higher than the 30HA 40NP samples, further highlighting the structural difference between CGs and HGs (Figure 6A). However, for the CGs, the NPs had a detrimental effect on the sample height, even in the swollen state.

The elastic moduli of all HGs were lower than 30 kPa, which ranks the HGs in the range of soft cutaneous tissue (Figure 6B) [43,44]. The highest elastic moduli, with around 20 kPa, were found for the 30HA and 30HA 40NP HGs, followed by the 10HA HG with 2.0–3.9 kPa, which is in line with previous results for the same type of HG [30,32]. The lowest elastic modulus was found for the CGs with 0.5 kPa. Loading the gels with NPs did not significantly influence the elastic modulus for any HG variant. The CGs were expected to have at least an elastic modulus in the range of the 10HA gel, due to the cryoconcentration effect before crosslinking [39]. To investigate the contradiction of the CG properties compared to those of the HGs—higher stability towards HYAL, but lower elastic modulus—the porosity of the 10HA and cryo 10HA gels was determined (Figure 6C). At the same mass, the CGs have a significantly higher porosity of 89.7+/−5.7% than the HGs, with 73.6+/−3.9%. Due to this, CGs generally have a lower material density, e.g., thinner pore walls, and, thus, a lower elastic modulus. On the other hand, the cryoconcentration increases the entanglement of the polymer chains and possibly the crosslinking, therefore, reducing the digestibility of the CGs by HYAL [41,42].

**Figure 5 jfb-14-00160-f005:**
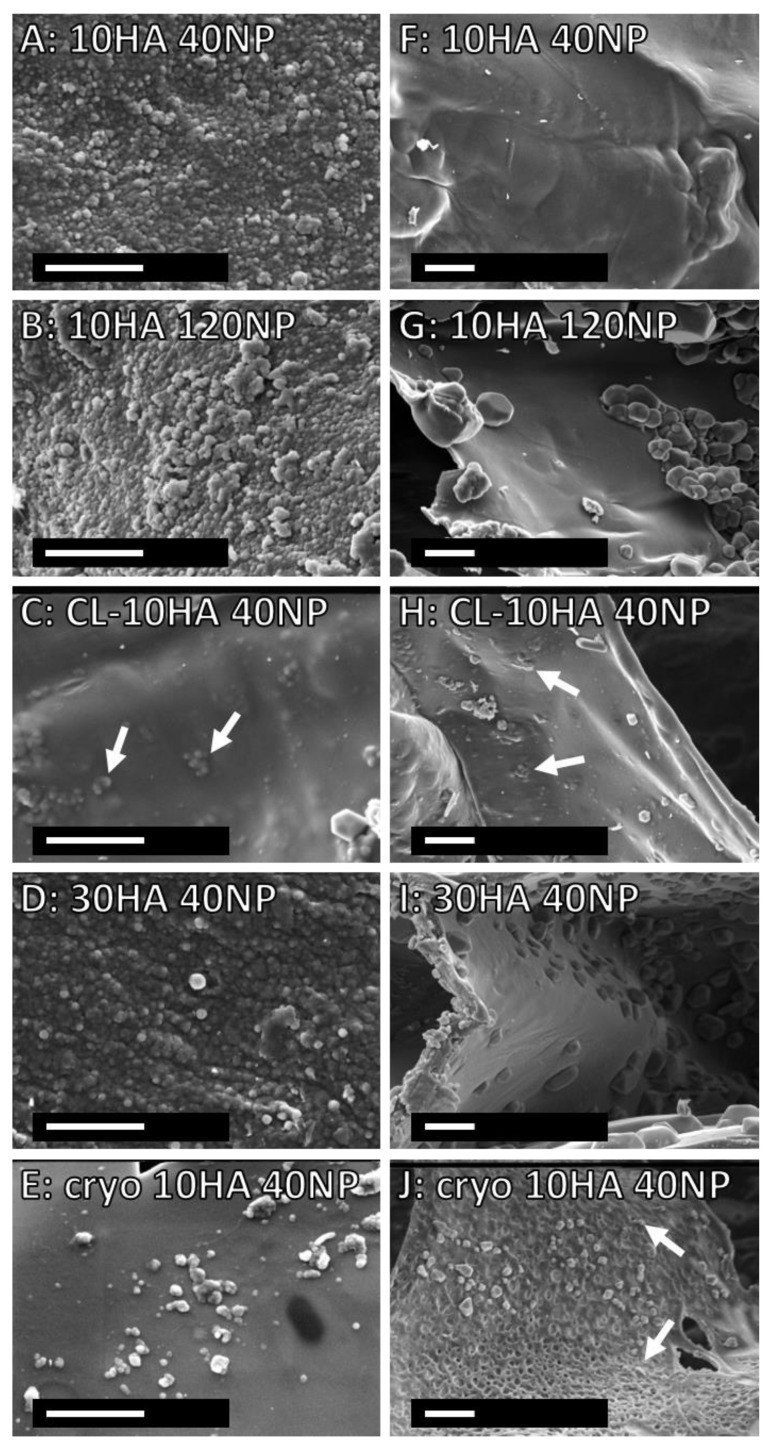
SEM images of HG and CG scaffolds after fabrication with NPs. (**A**–E) (magnification 40,000×): top view and (**F**–**J**) (magnification 20,000×): cross-section. The white arrows indicate NPs. Scale bar: 2 µm.

**Figure 6 jfb-14-00160-f006:**
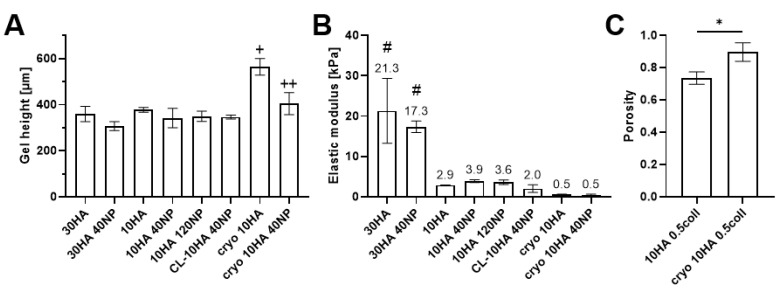
Gel height after swelling (**A**), elastic modulus (**B**), and porosity of HGs and CGs (**C**). (**A**,**B**) are measured on 3 mm punches of 50 µL disc-shaped gels after swelling in PBS at room temperature. (**C**) is determined on 200 µL cylindrical gels in PBS at room temperature. One-way ANOVA (**A**,**B**) and unpaired *t*-test (**C**): statistically significant for * *p* < 0.05, statistically significant difference: # against all 10HA HGs and CGs, + against all gels, ++ against 30HA 40NP and cryo 10HA, *n* = 3.

### 3.6. Proliferation and Viability of a Macrophage Cell Line

The proliferation of RAW264.7 on the HGs was studied through the assessment of the cell metabolic activity. As shown in Figure 7A, no significant differences were observed for NP-loaded HGs. Compared to 24 h, a significant increase (*p* < 0.05) in cell number was observed after 48 h for all gels. Moreover, a higher cell number was observed for CGs (cryo 10HA and cryo 10HA40NP) compared to HGs. This increased proliferation in the CGs can be related to their open pore structure that can promote cell growth and attachment to the gels (Appendix A, see I–K compared to A–H).

On the other hand, RAW264.7 proliferation was unchanged compared to control samples when treated with the HG extracts, demonstrating the biocompatibility of the supernatants for the cells (Appendix A).

Cell viability was evaluated by visualizing the presence of living and dead cells on gels after 48 h (Figure 7B). The quantitative analysis revealed no significant differences between unloaded and NP-loaded HGs, confirming that the introduction of the NPs in the HGs did not compromise the well-known biocompatibility of HA and coll [45,46,47]. Additionally, representative fluorescent microscopy images (Appendix A), showed the sparse appearance of dead compared to living RAW264.7 cells on top of the gels. 

### 3.7. Anti-Inflammatory Effect

Macrophages are key effectors in response to external material implantation, known as the foreign body reaction (FBR). Their direct contact with the material surface can induce the secretion of pro-inflammatory mediators. An extended unresolved inflammation then leads to the fusion of macrophages into foreign body giant cells (FBGCs) to phagocytose the material, inducing the recruitment of fibroblasts for fibrous encapsulation and scaffold failure [48,49]. During the initial phase of this FBR, pro-inflammatory M1-type macrophages lead to acute reactions to the implanted material, while anti-inflammatory M2-type macrophages control the resolution of inflammation and induce the subsequent healing stage [50].

In order to assess the presence of FBGC on the surface of the gels, the actin cytoskeleton and cell nuclei were visualized after 48 h of cell incubation, obtaining fluorescent images and quantitative measurements of macrophage surface area (Figure 8). In general, macrophages were predominantly round-shaped. However, HGs without NPs (30HA and 10HA) presented some multinucleated cells displaying a higher degree of spreading, and some were elongated with a spindle-shaped morphology. This morphology corresponds to activated pro-inflammatory M1-type macrophages. However, when NPs were loaded into the HGs, lesser multinucleated FBGC were observed, and smaller cell surface areas were measured (10HA 40NP, 10HA 120NP, and 30HA 40NP). For CL-10HA 40NP, this effect was slightly diminished. Importantly, CGs with and without NPs (e.g., cryo 10HA 40NP and cryo 10HA) showed a reduced number of FBGCs compared to HGs. 

During inflammation, multiple cellular inflammatory mediators, such as cytokines and chemokines, are produced by both immune and resident cells, creating a complex network of biochemical factors. These factors represent an important therapeutic target in the management of inflammatory diseases. In this sense, the anti-inflammatory effect of gels in LPS-stressed macrophages was evaluated by quantifying the release of representative inflammation-related factors when macrophages were seeded on the top of the gels. Several studies have shown that bacterial LPS induces macrophage polarization into a pro-inflammatory M1-type phenotype via binding to cellular Toll-like receptors. This binding leads to the activation of NF-κB signaling pathways, stimulating the release of pro-inflammatory mediators, such as NO, IL-1β, IL-6, and TNF-α [51].

Here, the levels of NO were measured (Figure 9A). Unstimulated macrophages produce a basal NO amount of 10% with respect to the LPS-stimulated group without any gel. The 10HA, CL-10HA 40NP, and cryo 10HA gels showed no changes in the NO production by macrophages compared to the control. Moreover, this factor was significantly increased (*p* < 0.05) for the 30HA HG. Macrophages experience phenotypic changes dependent on the molecular weight of HA; low molecular weight HA (<5 kDa) leads to a pro-inflammatory response, while high molecular weight HA (>800 kDa) leads to a pro-regenerating response [52,53]. The increment in NO production for 30 HA gels might be related to a higher release of low molecular weight HA fragments due to an increased HA concentration used for gel fabrication and subsequent cell-mediated enzymatic degradation. On the contrary, HGs containing NPs reduced the release of NO in all cases, except for CL-10HA 40NP. The highest reduction in NO production (75%) was achieved for the 10HA 120NP gel, demonstrating a concentration-dependent effect. Given that the NPs in the CL-10HA 40NP samples are hidden below the surface (Figure 5C,H), this hinders the access to and internalization of NPs by cells, thus, reducing their efficacy. 

Results on NO release were in good agreement with findings on macrophage area. Whereas gels without NPs showed a larger area and a higher NO release, gels with direct access to the NPs displayed reduced values for both (Figure 8B and Figure 9A). Importantly, the reduced number of FBGC and lowered NO levels alike confirm that the anti-inflammatory effect of the NPs is preserved when they are included in the gels. Of note, for these effects, a direct cell–NP contact is necessary.

However, treatment of the RAW264.7 cells with gel extracts did not modify the NO production (Appendix A), confirming the low release of NPs from the gels (Figure 3B). The situation in vivo might be different due to the possible enzymatic degradation of the gels by HYAL that could result in a release of the NPs, reducing NO production.

In order to substantiate the anti-inflammatory effect of 10HA and 10HA 120NP HGs, with the latter achieving the strongest reduction in NO production, the release of different pro- and anti-inflammatory mediators (TNF-α, IL-6, IL-10, IL-12p40, IL-23, and PGE_2_) was analyzed by ELISA (Figure 9B). When RAW264.7 cells seeded on 10HA were stimulated with LPS, an overproduction of all of these factors was evident. Here, the following mediator concentrations were determined: 11800, 2500, 1700, 360, and 220 pg/mL for PGE_2_, TNF-α, IL-6, IL-10, and IL-12 p40, respectively. A significant reduction was achieved for all factors when NPs were present in the HG (10HA 120NP), with 2520, 1580, 1460, 310, and 38 pg/mL for the same mediators. 

The reduction in cellular NO, TNF-α, and IL-12p40 levels when RAW264.7 cells were in contact with NP-loaded HGs is in agreement with previous results obtained with KT-based NPs [34]. In this previous work, a reduction in NO production was observed after 24 and 48 h of NP exposure. Moreover, real-time PCR revealed *IL12b* gene repression after 1 and 7 days and *Tnfa* gene repression after 1 day due to cell exposure to the NPs. In the present study, a significant reduction was observed as well after 24 h of exposure (Figure 9A). Likewise, a significant decrease in both TNF-α and IL-12p40 levels after 24 h on NP-containing HGs was detected (Figure 9B). On the other hand, a reduced anti-inflammatory IL-10 production by RAW264.7 cells was observed, opposing previous results indicating an induced *Il10* gene expression after 1 and 7 days of NP exposure [34]. In the present work, NPs are embedded in the HGs. Consequently, more sustained and controlled effects of the NPs are expected in comparison to the direct exposure of cells to the NPs. This could explain the decreased amount of IL-10 found here. Analogous to previous work, IL-23 cytokine levels were also determined. However, the cytokine levels were not high enough to be detectable by ELISA.

Furthermore, 10HA 120NP gels induced a significant reduction for two other important pro-inflammatory factors, i.e., PGE_2_ (*p* < 0.001) and IL-6 (*p* < 0.01). NSAIDs, such as KT, exert their activity mainly by inhibiting the action of COX enzymes involved in the biosynthesis of PG and thromboxane from arachidonic acid. PGE_2_ is one of the most abundant PG produced in the body, and its dysregulation has an important role in chronic inflammation [54]. In accordance with the KT mechanisms of action, PGE_2_ was the factor most significantly overexpressed by the LPS stimulus and reduced by the presence of the NPs in the HGs. 

In summary, these findings demonstrate the NPs’ suitability in enhancing the anti-inflammatory properties of HA/coll-based gels and, thus, the composite’s potential to act as an anti-inflammatory wound dressing. NP concentrations and loading technique for NP incorporation significantly influenced the efficacy of NPs, with higher NP concentrations being more effective. Importantly, direct contact of the cells with the NPs on the gel surface was mandatory for the effect. We recognize that our study has potential limitations, such as using a macrophage cell-line instead of primary cells, which should be additionally investigated in future studies. In order to enable a comparison with commercially available wound dressings, further studies should also consider analyzing NP-modified versions of those.

## 4. Conclusions

In this study, different HA/coll-based HGs and CGs containing KT NPs were designed, which displayed preparation-dependent NP distribution, surface topography, and elastic moduli. All gels that allowed direct contact of the cells with the NPs on their surface showed a reduction in the pro-inflammatory marker NO, while maintaining high cell viability and proliferation. In addition, by directly comparing 10HA 40NP and 10HA 120NP HG, concentration-dependent cellular effects of NPs were demonstrated. The 10HA 120NP gels also markedly reduced cell-related production of the pro-inflammatory markers PGE2 (to 21%), IL-12 p40 (to 17%), TNF-alpha (to 63%), and IL-6 (to 84%) in inflammatory conditions. The presented in vitro findings on HA/coll-based gels containing KT NPs demonstrate their potential for reducing pro-inflammatory macrophage polarization. This could provide a promising asset in treating chronic wounds via the immune-centered approach, in particular when combined with other anti-inflammatory agents, such as sHA. The in vivo relevance of these findings should be addressed in future studies utilizing an animal model for chronic skin wounds.

## Figures and Tables

**Figure 1 jfb-14-00160-f001:**
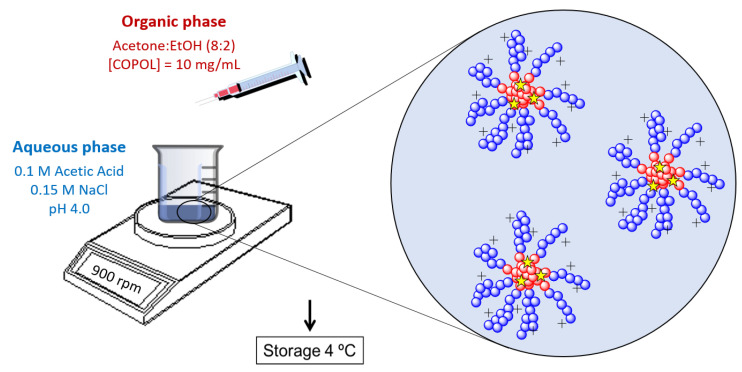
NP synthesis by nanoprecipitation method. Blue dots: positively charged poly(vinyl imidazole); red dots: hydrophobic ketoprofen derivative core; yellow stars: coumarin-6 dye or any other hydrophobic drug loaded to the hydrophobic core. COPOL: copolymer poly(HKT-co-VI) (48:52).

**Figure 2 jfb-14-00160-f002:**
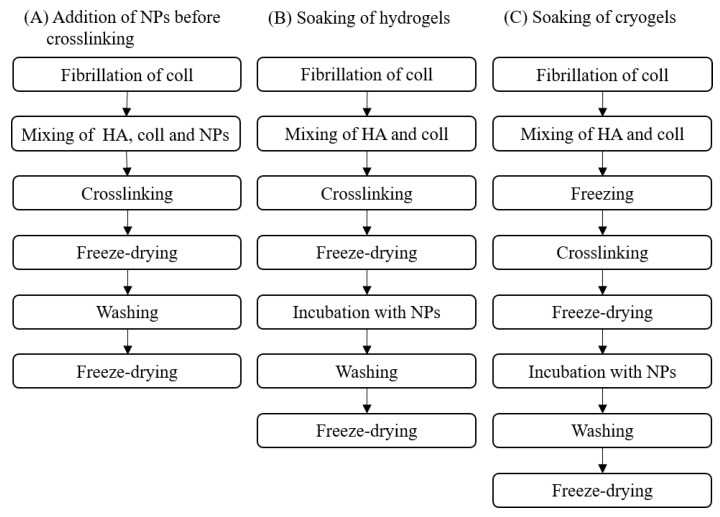
Different methods of NP incorporation during the gel fabrication process.

**Figure 3 jfb-14-00160-f003:**
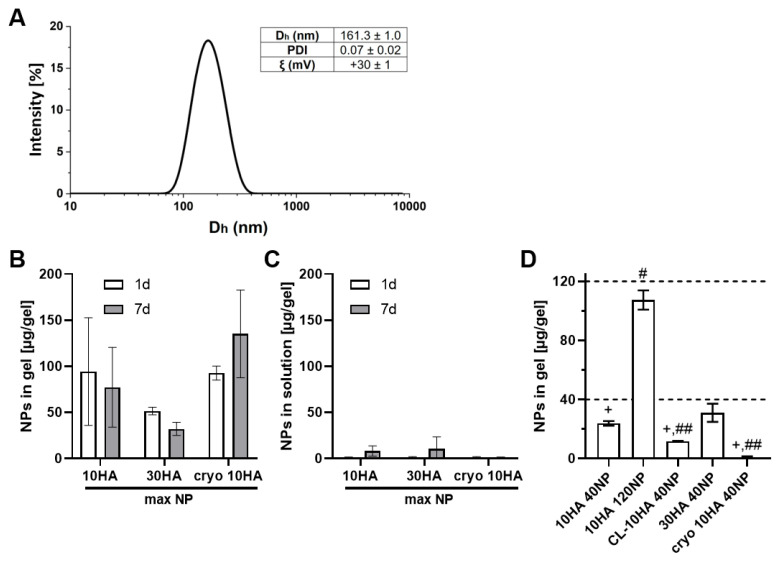
(**A**): Hydrodynamic properties of the KT NPs: hydrodynamic diameter (D_h_), polydispersity index (PDI) determined via dynamic light scattering, and zeta potential (ξ), as measured with laser Doppler electrophoresis. (**B**,**C**): Incubation of different gels containing NPs in PBS at 37 °C for 1 d and 7 d. NPs found after digestion of the gels (**B**) and in the incubation solution (**C**). Loading of gels was carried out with 250 µL of 1 mg/mL NP stock solution. Mood’s median test: no statistical significance found for *p* < 0.05; 10HA & 30HA *n* = 4; cryo 10HA *n* = 3. (**D**) Controlled loading of the gels with NPs. The graph shows the NP concentration in the different gels found after digestion as determined by fluorescence measurement. Statistics: *p* < 0.05 was considered statistically significant. One sample *t*-test: + statistically significant difference to the target concentration of 40 µg NP/gel; one-way ANOVA: # statistically significant difference against all other gels; ## significantly different to 10HA 40NP, 10HA 120NP and 30HA 40NP, *n* = 3.

**Figure 4 jfb-14-00160-f004:**
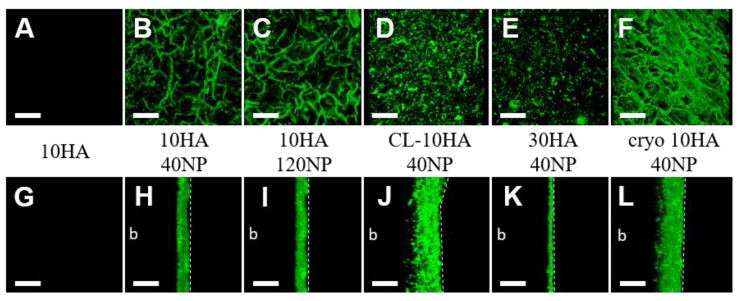
CLSM images of different gels containing C6-loaded NPs after swelling in PBS. Top view (row **A**–**F**) and cross-section (row **G**–**L**). Scale bar: 100 µm. 30HA and cryo 10HA did not show any fluorescence, as was the case for the 10HA samples (**A**,**G**). The dotted line reflects the top side of the gels, and b is the direction of the bottom side.

**Figure 7 jfb-14-00160-f007:**
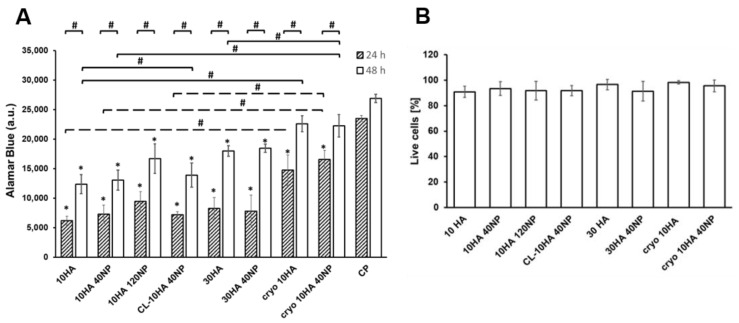
Cell proliferation after 24 h and 48 h (**A**), and cell viability after 48 h (**B**) on HGs and CGs. Data are represented as mean ± SD values. A: ANOVA performed between formulations and cells on culture plate (CP) (* *p* < 0.05), between 24 h and 48 h time points (# *p* < 0.05) between CGs and HGs (# *p* < 0.05) and between NP-loaded CGs and NP-loaded HGs (# *p* < 0.05); (*n* = 3).

**Figure 8 jfb-14-00160-f008:**
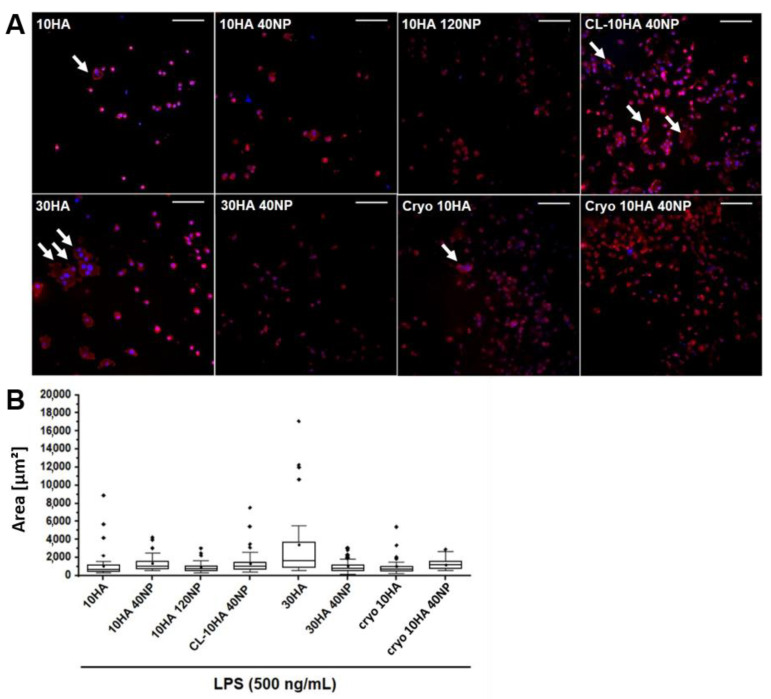
LPS-stimulated RAW264.7 cell spreading and multinucleated FBGC formation as evaluated by fluorescence microscopy. (**A**) Quantification of the area occupied by macrophages with Image J analysis software; white arrows indicate FBGC formed by the fusion of macrophages; scale: 200 µm. (**B**) Data are given in median (line), 25 and 75% quartiles (box) and 10th and 90th percentiles (whiskers). Data outside 10th and 90th percentile are marked with ◆.

**Figure 9 jfb-14-00160-f009:**
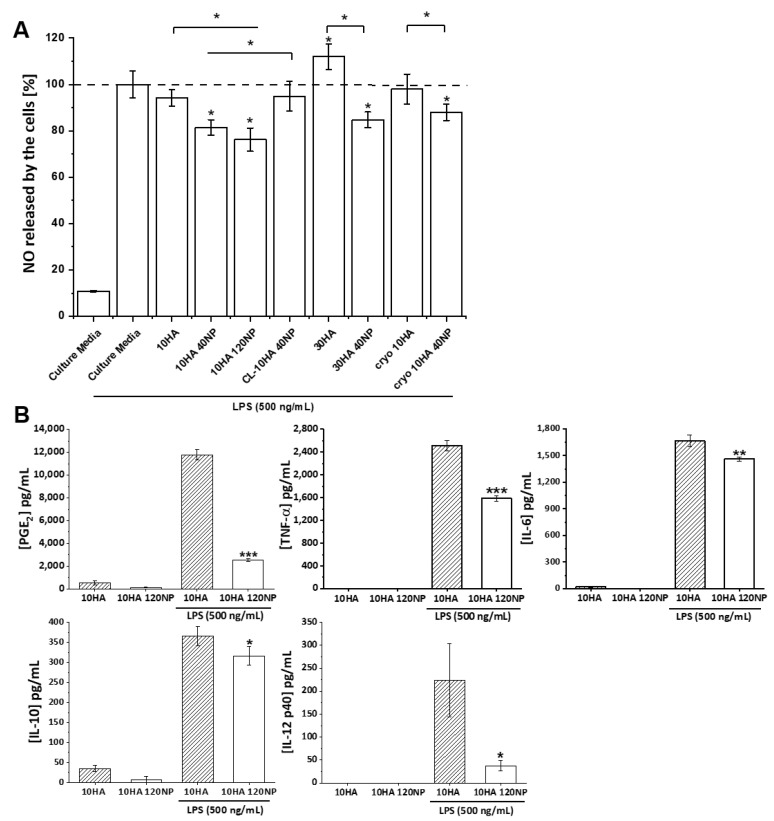
(**A**) Effect of gel type on NO production in LPS-stimulated RAW264.7 cells. Mean ± SD values are relative to control LPS-stimulated cells in culture media only, in which NO production was taken as 100%. ANOVA between HGs and control LPS-supplemented culture media w/o HGs or CGs (* *p* < 0.05) was performed; *n* = 4 (**B**) Effect of 10HA and 10HA 120NP on the release of the pro-inflammatory mediators PGE_2_, TNF-α, IL-6, IL-10, and IL-12 p40 in non-stimulated and LPS-stimulated RAW264.7 cells as determined by ELISA. For both tests, cells were seeded on top of the HGs for 24 h with subsequent incubation with LPS overnight. Data are represented as mean ± SD values. ANOVA for LPS-stimulated cells on 10HA and 10HA120NP HGs (* *p* < 0.05, ** *p* < 0.01 *** *p* < 0.001) was performed; *n* = 6.

**Table 1 jfb-14-00160-t001:** Fabricated HG variations with aspired NP concentration.

Label	c HA-MAC ^1^ [mg/mL]	c Coll ^1^ [mg/mL]	Gel Type	Loading Method	Target NP c [µg/HG]
Binding and release studies of NPs					
10HA max NP	10	0.5	HG	Soaking ^2^	-
30HA max NP	30	0.5	HG	Soaking ^2^	-
cryo 10HA max NP	10	0.5	CG	Soaking ^2^	-
Cell experiments and determination of the properties					
10HA	10	0.5	HG	-	-
10HA 40NP	10	0.5	HG	Soaking	40
10HA 120NP	10	0.5	HG	Soaking	120
CL-10HA 40NP	10	0.5	HG	Crosslinking	40 ^1^
30HA	30	0.5	HG	-	-
30HA 40NP	30	0.5	HG	Soaking	40
cryo 10HA	10	0.5	CG	-	-
cryo 10HA 40NP	10	0.5	CG	Soaking	40

^1^ Before addition of LAP; ^2^ 1 mg/mL NP solution.

**Table 2 jfb-14-00160-t002:** Digestion of gels in 600 µL buffer containing 1000 U/mL hyaluronidase (HYAL), 0.15 M NaCl and 0.01 M acetic acid with pH 5.35.

Gel	NP Content	Incubation Temperature [°C]	Incubation Time [d]	Change of Supernatant
10HA	Max NP, 40NP, and 120NP	37	1–2	-
30HA	Max NP and 40NP		4–6	2×
cryo 10HA	Max NP and 40NP		4–6	2×

## Data Availability

The raw data supporting this study’s findings are available upon reasonable request from the corresponding authors.

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
