# Peer review of "Ketoprofen-Based Polymer-Drug Nanoparticles Provide Anti-Inflammatory Properties to HA/Collagen Hydrogels"

_jfb, 2023, doi:10.3390/jfb14030160_

Round 1

Reviewer 1 Report

Respected authors

Many thanks for your efforts done during this valuable work.

Really I enjoyed reading this manuscript.

I have only one comment regarding the Figure of SEM, as the magnification should be indicated in the figure, also, contrast can be adjusted to be more clear and the required elements can be easily detected, as in Figure 1C, D, E and J, it is so difficult to check.

Author Response

We thank you very much for your quite positive review and your helpful comment. The contrast of all SEM images in the previous Figure 3 (now Figure 5) has been increased accordingly, improving clarity. Further, the magnification has been indicated in the figure description.

Reviewer 2 Report

Dear authors,

Congratulation for your intense work!

The results are very interesting, but I have some recommendations for you, before publication, in order to finally have a very good article.

1.     Line 102 – since you present in this sentence the aim of you study, I don’t find necessary the two references, or you should reformulate it. 

2.     Please add and emphasize the novelty/innovative aspects of this research work in the last paragraph of the introduction.

3.     Line 132 – C6, Sigma-Aldrich – you should also add the location of the manufacturer.

4.     Line 141 – Dynamic light scattering (DLS), please add the name of the used device and the manufacturer.

5.     I recommend to the authors to provide a figure/scheme representing also the nanoparticle synthesis and to move Figure S1 to the main manuscript, as it is very important for the article. 

6.     Please be careful on writing empirical formulas: Line 150 – Na2HPO4, correct is Na2HPO4. Line 243 – CO2 and so on.

7.     Line 157 – Please mention freeze-drying parameters (pressure, temperature, time, device manufacturer).

8.     Can you provide some data about the chemical structure of the NP and the hydro/cryogel?

9.     Line 207 – “rcf” should be written with capital letters: RCF. 

10.  2.8. Hydro- and cryogel morphology – please mention the working parameters: voltage, were the samples metallized?

11.   Figure 3. – Magnification should be also added.

Author Response

We thank Reviewer 2 very much for his/her detailed comments and assistance to further improve the manuscript. We have made the following changes:

1) The sentence on cell compatibility of HA/coll-based HG and associated references was moved to another appropriate location in the manuscript (line 79). The first part of the paragraph was reformulated.

2) The novelty of the study was additionally emphasized. This is the first known study on a combination of KT NP with HA/coll-based hydro- and cryogels, addressing the physicochemical and anti-inflammatory properties of the composite materials. 

3) The location and country of the supplier for coumarin-6 was added.

4) The name and manufacturer of the DLS device as well as its equipment was added.

5) A figure on NP synthesis was added to the methods section (Figure 1). Figure S1 was moved to the main manuscript (now Figure 2).

6) Many thanks for the careful reading. We have corrected the chemical formulas.

7) The manuscript (2.4.1) now contains detailed information on the lyophilization system and the used lyophilization parameters.

8) The chemical structure of the poly(HKT-co-VI)(48:52) copolymer, the description of the microstructure of the copolymer as well as the extensive characterization of the NP hydrodynamic properties have been described  in the provided citation [34] Pharmaceutics 2020, 12, 723, Before nanoprecipitation, the copolymer was investigated by 1H-NMR. The information about this has been added.

9) We have implemented this suggestion and changed “rcf” to “RCF”.

10) We have included the working parameters for SEM. The samples were not metallized but coated with carbon as indicated in the method section.

11) The magnification was added to the figure description accordingly.

Reviewer 3 Report

This is a very nice study. The introduction is very thorough. Enough data has been presented to support the argument, and the story's logic is sound. 
I only have one request from the authors and this is to improve the quality of Figure 1.

Author Response

We thank you very much for your quite positive review. Figure 1 (now Figure 3) has been adapted by enlarging Figure 3A and rearranging Figure 3D.

Finally, we performed a spell check on our manuscript and removed minor spelling errors.

Round 2

Reviewer 2 Report

Congratulations for your work!

The manuscript can be published in the present form!